# The Effect of Low Glycemic Index Mediterranean Diet and Combined Exercise Program on Metabolic-Associated Fatty Liver Disease: A Joint Modeling Approach

**DOI:** 10.3390/jcm11154339

**Published:** 2022-07-26

**Authors:** Ritanna Curci, Antonella Bianco, Isabella Franco, Angelo Campanella, Antonella Mirizzi, Caterina Bonfiglio, Paolo Sorino, Fabio Fucilli, Giuseppe Di Giovanni, Nicola Giampaolo, Pasqua Letizia Pesole, Alberto Ruben Osella

**Affiliations:** 1Laboratory of Epidemiology and Statistics, National Institute of Gastroenterology, IRCCS, “S. de Bellis” Research Hospital, 70013 Castellana Grotte, BA, Italy; ritanna.curci@irccsdebellis.it (R.C.); antonella.bianco@irccsdebellis.it (A.B.); isabella.franco@irccsdebellis.it (I.F.); angelo.campanella@irccsdebellis.it (A.C.); dottoressamirizzi@yahoo.com (A.M.); catia.bonfiglio@irccsdebellis.it (C.B.); 2Department of Electrical and Information Engineering, Polytechnic of Bari, 70126 Bari, BA, Italy; paolo.sorino@poliba.it; 3Department of Radiology, National Institute of Gastroenterology, IRCCS, “S. de Bellis” Research Hospital, 70013 Castellana Grotte, BA, Italy; fabio.fucilli@irccsdebellis.it (F.F.); giuseppe.digiovanni@irccsdebellis.it (G.D.G.); nicola.giampaolo@irccsdebellis.it (N.G.); 4Laboratory of Clinical Pathology, National Institute of Gastroenterology, IRCCS, “S. de Bellis” Research Hospital, 70013 Castellana Grotte, BA, Italy; letizia.pesole@irccsdebellis.it

**Keywords:** lifestyle, fatty liver, metabolic dysfunction, combined exercise program, joint modelling

## Abstract

Background: Excessive caloric intake and reduced energy expenditure are associated with the onset of metabolic-associated fatty liver disease (MAFLD). The aim of this study was to probe the benefits of a low glycemic index Mediterranean diet (LGIMD) and a combined exercise program (CEP) on MAFLD by monitoring the clinical process through anthropometric measurement, body mass index (BMI), and specific biomarkers, such as the Homeostatic Model Assessment for Insulin Resistance (HOMA-IR). Methods: The study was conducted at the National Institute of Gastroenterology, ‘S. de Bellis’, Italy. Subjects were invited to join the study for 12 months. Results: 54 participants were enrolled. Joint modeling of longitudinal and time-to-event data was applied. Overall, a statistically significant direct effect of LGIMD/CEP adherence on ln (BMI), a statistically significant direct effect of LGIMD/CEP adherence on time-to-event and a strong statistically significant direct effect of log (BMI) on time-to-event were observed. In addition, a statistically significant direct effect of LGIMD/CEP adherence on ln(HOMA-IR), a statistically significant direct effect of LGIMD/CEP adherence on time-to-event and a statistically significant direct effect of ln(HOMA-IR) on time-to-event were observed. Conclusions: LGIMD/CEP significantly improved MAFLD status; in addition, longitudinal BMI and HOMA-IR were good predictors of the disappearance of diagnostic criteria for MAFLD.

## 1. Introduction

Metabolic Associated Fatty Liver Disease (MAFLD) affects about a quarter of the world’s adult population and constitutes an important burden on all health systems. In 2020, an international consensus proposed the new inclusive concept of MAFLD as a convenient definition for identifying patients with fatty liver disease [1]. MAFLD encompasses a spectrum of risk factors that comprises simple steatosis plus metabolic dysfunctions [2]. Differences between Non-Alcoholic Fatty Liver Disease (NAFLD) and the MAFLD has now been evidenced and the MAFLD definition has been well-established [3]. In the absence of approved specific pharmacological therapies, lifestyle interventions remain a good choice for the treatment of MAFLD [4]. The Mediterranean diet (MD) appears to be the best choice of eating behavior [5,6], but until now, only a limited number of studies have evaluated the association between diet and MAFLD. A recent study has highlighted the association between healthy dietary patterns and hepatic steatosis associated with metabolic dysfunction [7]. However, some aspects of this relationship warrant further study. Furthermore, the independent role of exercise in the treatment of MAFLD remains unclear. A recent review of patients with established MAFLD reported that both aerobic and resistance exercise training, even without significant weight loss, produces a 20–30% reduction in intrahepatic lipid content as assessed by non-invasive methodologies [8]. Other studies have been focused exclusively on verifying the benefits of aerobic-type exercise programs (EP) on MAFLD, confirming their effectiveness [9,10]. The effectiveness on MAFLD of combined exercise, involving a planned, structured, and repetitive set of movements, is to this point unclear [11]. In fact, there is a general agreement that additional research is needed to determine which type of physical activity is the most effective for MAFLD while following the FITT principles (Frequency, Intensity, Time and Type) [12]. The joint modeling of longitudinal and survival data has received attention in the literature, but no software to implement it was available until recently. In this type of model, an association, pointed out by shared random effects, between a longitudinal process and the time-to-event, is assumed [13]. A framework to assess the predictive ability of a biomarker (or another type of measure) on time-to-event can effectively be gained using this methodology, as it can assess the impact that a longitudinal variable, measured with error, has on the time to an event of interest. Moreover, it reduces bias and improves precision as compared to simpler approaches [14]. We hypothesized that MAFLD patients could benefit from following a diet and a combined aerobic and resistance EP, and that measuring the Body Mass Index (BMI) and/or applying the Homeostatic Model Assessment for Insulin Resistance (HOMA-IR), may constitute a useful tool, allowing the easy monitoring of the MAFLD clinical treatment process.

This observational study was aimed at probing the impact of adherence to a 12-month Low Glycemic Index Mediterranean Diet (LGIMD) and a CEP on the MAFLD clinical course. We focused on two aspects, an anthropometric measurement, the BMI and a biomarker, the HOMA-IR Index, to monitor longitudinal processes and a time-to-event clinical outcome process (disappearance of MAFLD diagnostic criteria). We assumed that these parameters can effectively assess the impact that longitudinal variables have on the time to an event of interest. In addition, this study offered an opportunity to assess the predictive ability of the BMI and the HOMA-IR Index.

## 2. Materials and Methods

### 2.1. Participants

The study was conducted by the Laboratory of Epidemiology and Statistics of the National Institute of Gastroenterology, “S. de Bellis” Research Hospital, Castellana Grotte (BA), Italy, from March 2018 to February 2020. Enrolled subjects were invited to follow a 12-month diet and CEP. Patients were recruited over two years, for a total of 54 participants, assigned to non-contemporary groups.

### 2.2. Study Design

A convenience sample was recruited for this observational study. General practitioners were invited to send patients suffering from metabolic dysfunctions. The inclusion criteria were: age between 18 and 65 years plus Metabolic dysfunctions [15,16,17]. Exclusion criteria comprised: significant orthopedic or neuromuscular limitations; overt cardiovascular disease and revascularization procedures; stroke; clinical peripheral artery disease; any severe medical condition that could prevent the subject from following a nutritional program, and inability to follow a LGIMD for any reason.

### 2.3. Data Collection

Participants signed informed consent and the study was conducted in accordance with the Helsinki Declaration and approved by the local Ethical Committee. Participants completed a structured questionnaire about sociodemographic aspects, medical history, and lifestyle. The European Prospective Investigation into Cancer and Nutrition Food Frequency Questionnaire (EPIC FFQ) was used to probe eating behavior [18]. Blood samples were collected after overnight fasting, and biochemical measurements were performed using standardized methods. Anthropometric measurements (weight, height, waist circumference) were taken by trained dietitians. Weight measurements were taken with SECA mechanical scales (model 700; Hamburg, Germany), whereas height was measured with a wall altimeter (model 206; 220 cm; SECA, Hamburg, Germany). Subjects wore only underwear during measurement. The presence of hepatic steatosis was assessed and categorized (0 absent, 1 mild, 2 moderate, and 3 severe) using an Esaote MyLab70 XVG device and a Convex 5-Mhz probe, following standard criteria [2]. All measurements, including Liver Ultrasound (LU), were performed at baseline and every three months.

### 2.4. Dietary Program

The Low-Glycemic Index Mediterranean Diet (LGIMD) was provided in brochure format, with graphic explanations organized according to a traffic light system [19] with a list of foods that could be eaten frequently (green foods), sometimes (yellow foods), and never (red foods). There were no calorie restrictions. The brochure also contained a daily dietary record, where participants indicated the code of each food consumed at breakfast, lunch, dinner, and during snacks. The diet’s composition is shown in Appendix A [20].

### 2.5. Exercise Program

The combined training protocol started with a 12-session conditioning period in which subjects perform low-intensity aerobic exercises (50–55% maximum Heart Rate (HRmax), accompanied by easy-to-perform strengthening exercises. To determine the maximum age-predicted heart rate, we used Tanaka’s formulas [21]. After this first period of conditioning, the training protocol was adapted specifically to considering the participants’ metabolic dysfunctions conditions. The EP structure is shown in Table 1.

Each exercise step lasted for 12 sessions; all participants wore a heart rate monitor. The EP was subdivided into three categories based on level of difficulty. The basic level included three sets of exercises consisting of 10 repetitions; the intermediate level, three to four sets of 15 repetitions; and the advanced level, which consisted of four to five sets of 20 repetitions. The aerobic exercise included treadmill walking, cycling, cross-training, and rowing. The strengthening exercises included free-body parts with the help of small tools and isotonic machines.

### 2.6. Adherence to LGIMD

The Mediterranean Adequacy Index (MAI) [22], was used to assess subjects’ compliance. We chose random weeks of the year, namely the 2nd, 7th, 10th, 16th, 23rd, 24th, 34th, 35th, 36th and 43rd. The participants were unaware of the days and weeks in which they would be evaluated. Compliance was defined as positive if the subject’s MAI was equal to or above the median expected value for the whole population under study; a median value of 7.5 with an inter-quantile range (IQR) of 5.4 was expected [22].

### 2.7. Adherence to the Exercise Program

To assess adherence to the CEP, the working year was split into 4 quarters. Adherence was assessed on the basis of frequency (at least 2 weekly attendances at training for a monthly total of 8, i.e., 24 attendances in three months), intensity (heart rate monitoring, based on the established work program; see Table 1) and workload (based on the established level of difficulty; see Table 1).

### 2.8. Statistical Analysis

We classified MAFLD participants into three subtypes based on the presence of different metabolic dysfunctions: hepatic steatosis in overweight/obese subjects (MAFLD1), hepatic steatosis and at least 2 metabolic abnormalities (MAFLD2), hepatic steatosis and type 2 diabetes mellitus (MAFLD3). A data description was performed by means (±SD), median (inter-quantile range), and frequencies (%) as appropriate. We defined as adherent those participants who remained above the expected MAI median for seven months and attended 60% of all sessions of CEP. An appropriate variable (LGIMD/CEP) was built. Joint modeling of longitudinal (BMI and HOMA-IR, which were repeatedly monitored over time) and time-to-event (disappearance of MAFLD diagnostic criteria) data was performed to estimate the effect of adherence to LGIMD/CEP on these processes. The two analyses were performed separately to clearly identify the predictive ability of BMI and HOMA-IR. Linear mixed models were performed for longitudinal trajectories of BMI and HOMA-IR over time. These variables were log-transformed to achieve normality. A random intercepts and linear longitudinal trajectory were set in the two models. In addition, an unstructured covariance matrix was applied. A Flexible Parametric Survival Model Cox with two degrees of was chosen [23]. The endpoint for the time-to-event model was the disappearance of the diagnostic criteria for MAFLD (mostly the absence of Hepatic Steatosis), withdrawal from the study or end of follow-up. All analyses were sequentially adjusted for Adherence to the LGIMD/CEP, Waist to Hip Ratio (WHR), Sex, PCR, ALT, GPT, GGT and BMI or HOMA-IR. To choose the best set of covariates, biological and statistical criteria (Akaike’s and Schwarz’s Bayesian Information Criteria) were used. A non-linear combination of coefficients was also used to test the overall effect of following the two programs on the log of BMI/HOMA-IR and on the time-to-event. Post-estimation tools were applied to obtain predictions. Statistical significance was set at *p* < 0.05. Stata 17.0 statistical software (StataCorp, 4905 Lakeway Drive, College Station, TX, USA) was used.

## 3. Results

Characteristics of the sample are shown in Table 2.

Fifty-four subjects, 41 with MAFLD, were included in the study. Of these, 74% had MAFLD1 and 28% had MAFLD3; many subjects met the requirements for the different MAFLD subtypes. Most of the MALFD participants were men, especially in MAFLD3. The mean age was 53 ± 10 years. At baseline, mean weight was 87 ± 16 kg, BMI 32 ± 5 kg/m^2^ and WHR index, 0.94. These values were higher in the MAFLD3 subtype (mean age 56 ± 8, mean weight 95 ± 16, mean BMI 34 ± 5 and mean WHR 0.98).

Appendix A shows the changes in MAFLD diagnostic criteria over time. These tended to decrease progressively until day 270, and then increase without reaching baseline values. These improvements were greater among adherent participants. In the whole sample, glucose metabolism percentages decreased progressively until the end of the project. In the first 7 months, 74% of participants exhibited a greater adherence to both LGIMD and CEP (about 70%). From the seventh month to the end of follow-up, adherence tended to decrease. The temporal evolution of BMI as well as some liver and metabolic biomarkers are shown in Appendix A. A trend to improvement is seen for all markers until the 270th day overlapping adherence to the LGIMD/CEP. In particular, glycemic control was optimal until the 270th day of the program.

### 3.1. BMI Longitudinal Trajectory and Time-to-Event

The results of joint modelling of longitudinal and time-to-event data are shown in Table 3.

Overall, there was a non-statistically significant direct effect of LGIMD/CEP on ln (BMI) (0.04, 95%CI −0.05; 0.13), a direct statistically significant effect of LGIMD/CEP on time-to-event (−1.78, 95%CI −3.40; −0.16), and a strong statistically significant direct effect of log (BMI) on time-to-event (−9.66, 95%CI −13.62; −5.69). The same behavior was observed in MAFLD subtypes.

### 3.2. HOMA-IR Longitudinal Trajectory and Time-to-Event

The results of joint modelling of longitudinal and time-to-event data are shown in Table 4.

Overall, there was a non-statistically significant direct effect of LGIMD/CEP on ln(HOMA-IR) (−0.27, 95%CI −0.61; 0.07), a direct statistically significant effect of LGIMD/CEP on time-to-event (−2.30. 95%CI −4.25; −0.34), and a statistically significant direct effect of ln(HOMA-IR) on time-to-event) (−3.38, 95%CI −6.69; −0.06). The same behavior was observed in MAFLD subtypes. The predicted hazard rates (speed to an event) over time are graphically displayed in Figure 1 (BMI) and Figure 2 (HOMA-IR).

As expected, the speed of time-to-event increases until the ninth month and then decreases.

Overall, there was a clear trend to MAFLD’s diagnostic criteria disappearance in the whole sample with an acceleration until the ninth month for both BMI and HOMA-IR in the whole sample as well as among adherent participants.

Both figures showed a stronger and faster disappearance of the diagnostic criteria in the subgroup with MAFLD3 than in MAFLD1, especially among adherent subjects.

The effect is more intense when HOMA-IR is considered. In fact, the difference between MAFLD1 and MAFLD3 is reduced in relation to BMI.

It is clear that participants who had followed both diet and exercise plans achieved stronger and faster benefits than the others.

## 4. Discussion

This study has shown that, following an LGIMD and a CEP, according to standard FITT principles, can induce changes that lead to the disappearance of some MAFLD diagnostic criteria. This event occurred both in the MAFLD group and MAFLD subtypes, but at a different speed. Moreover, the joint modeling of longitudinal (BMI and HOMA-IR) and time-to-event data proved to be a useful tool to monitor the underlying MAFLD clinical process.

For the management of NAFLD, numerous studies have demonstrated the effectiveness of lifestyle changes in the form of diet and physical activity, although the literature about lifestyle MAFLD treatment is scarce [9].

Recently, a practical guide for lifestyle modification using diet and exercise in NAFLD has been published. Reducing body weight induces a decrease in liver fat and improves glucose control and insulin sensitivity [24]. Moreover, exercise can significantly reduce hepatic steatosis even in the absence of considerable weight loss [9].

To achieve weight loss, several recommendations have been formulated. The Mediterranean diet, or a low-fat diet with a low intake of red and processed meat and commercially produced fructose, has been promoted [25]. Our LGIMD satisfies these requirements and has shown to be effective in NAFLD patients [19].

Furthermore, aerobic exercise has shown a positive effect on hepatic steatosis if performed as suggested in the literature; resistance training can be complementary to aerobic exercise but is not a substitute [8]. In this sense, our studies of subjects with NAFLD have shown that LGIMD and exercise, administered individually or in combination, produced positive effects [19,26,27].

In this study, the effects of changes in lifestyle were evident and stronger in adherent subjects to the program until at least 270 days from the beginning of follow up. These results are in line with a recent expert review [28]. In fact, in our MAFLD sample, metabolic dysfunctions showed an evident improvement, and then were followed by a tendency to return to initial values. However, none of the subjects dropped back to the baseline levels.

The difficulties in implementing and maintaining lifestyle changes for a long time are known, even if individuals are aware of the importance of a healthier lifestyle [29,30].

As shown in other studies, our data confirm that weight loss was associated with the disappearance of MAFLD diagnostic criteria [10,31]. This association was statistically significant in MAFLD subtypes, especially in MAFLD1. Many studies have confirmed a positive association between the BMI change and the change in liver fat content [31,32]; in fact, for every 1% decrease in body weight, there was a 1% decrease in liver fat content. The higher the initial BMI, the greater the reduction [33].

It has been shown that a low-saturated fat, low-GI diet, such as LGIMD, decreases hepatic fat in both sexes and reduces fat synthesis [19].

It is known that even small decreases in liver fat can be clinically relevant in subjects with high liver fat [34], especially if the diet can be maintained for a longer period.

Furthermore, the efficacy of the proposed CEP in the management of MAFLD is confirmed by numerous studies showing that exercise, especially aerobic exercise, is able to reduce visceral adiposity [9], thereby decreasing the influx of free fatty acids into the liver [35], and the systemic inflammation associated with metabolism. Moreover, it not only reduces the amount of adipose tissue, but also changes the structure and function of the adipocyte [9].

In addition, some studies showed that a CEP has an important effect on parameters such as waist circumference, body fat percentage, muscle mass, blood pressure, and resting blood pressure [36].

With regard to metabolic dysfunction improvement, aerobic exercise encompasses a series of benefits [37,38]. Specifically, aerobic exercise has an effect in reducing glycated hemoglobin (HbA1c) and serum cholesterol levels, and resting blood pressure [38], whereas resistance exercise was negatively associated with dyslipidemia, hypertension and insulin resistance [37]. Moreover, resistance exercise is well tolerated by subjects with a low cardiorespiratory capacity or who are overweight and cannot tolerate aerobic exercise.

Although less than in other groups, weight loss also occurred in the MAFLD3 subtype. As suggested [39], a loss of 5–10% of body weight in diabetic subjects could improve overall fitness, cardiovascular disease risk factors, and reduce HbA1c levels and the use of antihyperglycemic drugs.

In addition, subjects with MAFLD3 appeared to have a greater and faster reduction in MAFLD diagnostic criteria than the others, probably due to the numerous benefits of diet and exercise on glucose metabolism.

Studies show that in diabetic patients, adherence to the Mediterranean diet is associated with lower HbA1c levels and numerous other benefits, such as an improvement in fasting glucose homeostasis, insulin levels and a better insulin resistance index (HOMA) in both normoglycemic subjects and diabetic participants. Subjects with a high Mediterranean diet adherence score had a 15% reduction in glucose and insulin at baseline and a 27% increase in the HOMA index [40,41].

On the other hand, exercise and thus muscle contraction allow for an increase in glucose uptake by muscle through the displacement of GLUT-4 glucose receptors in the cell membrane independent of the action of insulin [42]. Also, exercise increases the muscle glucose storage as glycogen [43], it redirects circulating free fatty acids from the liver to the muscle [44], and it increases their uptake and oxidation by the muscle [45]. In addition, in the post-exercise period, glucose oxidation decreases at the expense of fat oxidation in order to replenish the glycogen store [46].

The observed effect of HOMA-IR was associated with the disappearance of MAFLD diagnostic criteria in all MAFLD subtypes. An insulin resistance reduction has been associated to a reduced progression of MAFLD and hepatic fibrosis [47]. As is well known, diabetes and hepatic steatosis share several molecular biology mechanisms, the most important one being insulin resistance [48]. HOMA-IR has been reported to decrease significantly with increasing exercise levels [49], and moderate-intense aerobic EP produced positive effects on glucose tolerance, independently of changes in abdominal fat. Even in adult subjects with abdominal obesity, the relationship between HOMA-IR and exercise is very strong, although this association may be influenced by waist circumference. It is interesting to note that insulin resistance appears long before diabetes [50] and seems to accelerate hepatic fibrosis. Therefore, HOMA-IR monitoring can play an essential role in preventing the late stages of hepatic steatosis [51].

The application of these new statistical methodologies to measurements that are routinely collected in clinical practice may identify useful tools for monitoring clinical processes that would otherwise require complex and costly laboratory research, saving time and avoiding the overburdening of the healthcare system. Moreover, the lifestyle treatment could reduce the recourse to pharmacological therapy, reducing costs further.

This study has strengths and limitations. The LGIMD and the CEP were administered under the supervision of specialized personnel. The CEP was built following specific FITT parameters and then adapted to each subjects’ MAFLD-associated conditions. Both LGIMD and CEP adherence was measured and an overall measure was estimated. Because we enrolled a convenience sample, it is possible that we could have recruited the most health-conscious people, so selection bias may be present. However, all diagnostic and follow-up assessments were performed in the same way, avoiding misclassification bias.

Statistical and biologic criteria were applied to obtain valid estimates while adjusting for all measured variables. However, residual confounding may still be present. Finally, in an observational study setting it is not possible to generalize the conclusions. Thus, randomized clinical trials are needed to disentangle the association among mechanisms underlying subtypes of MAFLD, diet and CEP. Studies of CEP and MAFLD should follow FITT recommendations to enhance the comparability of results with the aim of defining disease-specific protocols.

## 5. Conclusions

LGIMD and a CEP for 12 months with a progressively increased EP intensity up to 70–75% of HRmax, achieving a weekly volume of at least 180 min, lead to a significant improvement of MAFLD. Moreover, BMI and HOMA-IR, longitudinally measured, showed a good predictive ability for estimating the probability of time-to-event of interest (disappearance of MAFLD diagnostic criteria) in this sample. This type of methodology may help in the clinical-decision making process in order to reduce the resulting public health costs.

Therefore, together with specific drug therapy, a multidisciplinary team is needed to improve the management and treatment of MAFLD and to increase the motivation of patients to follow the correct lifestyle.

## Figures and Tables

**Figure 1 jcm-11-04339-f001:**
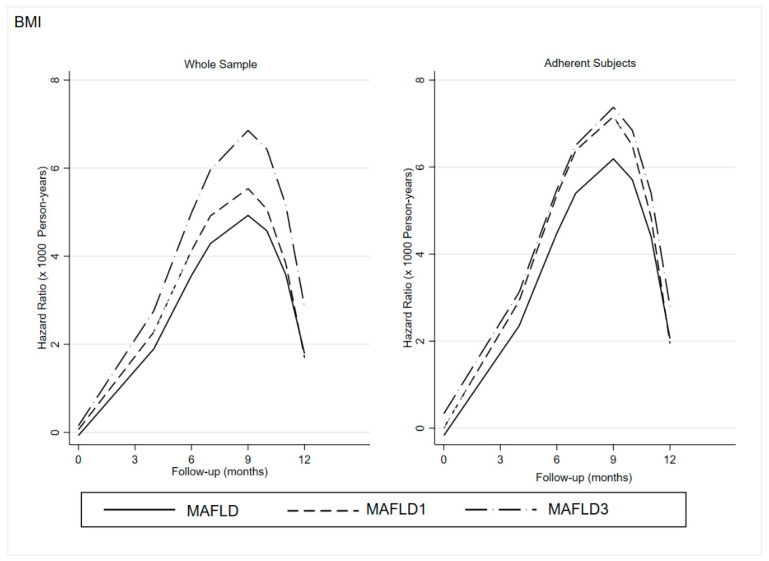
BMI: Effect of Adherence to LGIMD and CEP on Hazard Ratio of Time-to-Event (Loss of MAFLD Diagnostic Criteria for Whole Sample and Adherent Participants). BMI: Body Max Index; MAFLD: Metabolic-Associated Fatty Liver Disease: MAFLD1; presence of hepatic steatosis in overweight/obese subjects; MAFLD3: presence of hepatic steatosis and type 2 diabetes mellitus.

**Figure 2 jcm-11-04339-f002:**
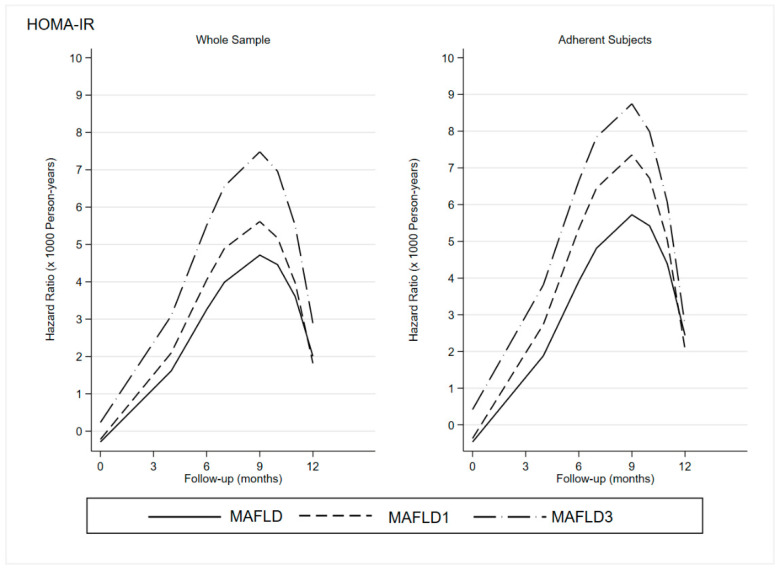
HOMA-IR: Effect of Adherence to LGIMD and CEP on Hazard Ratio of Time-to-Event (Loss of MAFLD Diagnostic Criteria for the Whole Sample and Adherent Participants). HOMA-IR: Homeostatic Model Assessment for Insulin Resistance; MAFLD: Metabolic-Associated Fatty Liver Disease: MAFLD1; presence of hepatic steatosis in overweight/obese subjects; MAFLD3: presence of hepatic steatosis and type 2 diabetes mellitus.

**Table 1 jcm-11-04339-t001:** Training protocol with progressive changes in duration and intensity.

Period	Intensity
Conditioning	Intensity at 50–55% HRmax + exercises at basic level
1st Step	5% increase in intensity (55/60% HRmax) + introduction of intermediate level exercises
2nd Step	Increased duration of aerobic work of 5′ (tot.15′) + reduced conditioning
3rd Step	5% increase in intensity (60/65% HRmax) and maintenance of intermediate level exercises
4th Step	Increased duration of 5′of the strengthening work (tot.15′) + reduction of cool down time
5th Step	5% increase in intensity (65/70% HRmax) + intermediate and advanced level exercises
6th Step	Increase duration of 5′of aerobic work (tot.20′)
7th Step	5% increase in intensity (70/75% HRmax) + advanced level exercises
8th Step	Increased duration of 5′ of the strengthening work (tot.20′) + reduction of all the other phases (aerobic excluded)

HRmax: Maximum Heart Rate.

**Table 2 jcm-11-04339-t002:** Characteristics of participants with MAFLD and subtypes of MAFLD.

	Whole Sample	MAFLD	MAFLD1	MAFLD3
N	54	41(75%)	41(75%)	15(28%)
Age	53.57 (10.21)	52.72 (11.18)	52.72 (11.18)	56.47 (8.18)
Sex				
Female	30 (56%)	19 (46%)	19 (46%)	6 (40%)
Male	24 (44%)	22 (54%)	22 (54%)	9 (60%)
SBP (mmHg)	130.37 (11.93)	131.95 (12.44)	131.95 (12.44)	134.00 (12.85)
DBP (mmHg)	82.22 (8.22)	82.93 (8.51)	82.93 (8.51)	82.00 (10.99)
Weight (kg)	87.19 (16.09)	90.51 (14.61)	90.51 (14.61)	95.94 (16.94)
BMI (kg/m^2^)	32.00 (5.14)	32.82 (4.96)	32.82 (4.96)	34.08 (5.94)
TGL (mmol/L)	1.74 (1.01)	1.78 (0.96)	1.78 (0.96)	1.70 (0.76)
Chol-T (mmol/L)	5.36 (1.01)	5.21 (0.94)	5.21 (0.94)	4.95 (0.87)
HDL-C (mmol/L)	1.15 (0.29)	1.14 (0.30)	1.14 (0.30)	1.12 (0.22)
LDL-C (mmol/L)	3.48 (0.61)	3.22 (0.48)	3.22 (0.48)	3.37 (0.42)
RC (mmol/L)	0.80 (0.46)	0.82 (0.44)	0.82 (0.44)	0.78 (0.35)
Glucose (mmol/L)	5.88 (1.57)	6.03 (1.68)	6.03 (1.68)	7.13 (2.33)
ALT (μkat/L)	0.54 (0.28)	0.59 (0.30)	0.59 (0.30)	0.59 (0.30)
Homa-IR	3.83 (2.34)	4.38 (2.37)	4.38 (2.37)	5.22 (2.12)
Insulin (pmol/L)	102.9 (66.21)	116.59 (68.98)	116.59 (68.98)	125.71 (78.78)
Waist (cm)	101.75 (11.84)	104.73 (10.75)	104.73 (10.75)	107.90 (10.54)
WHR (cm)	0.94 (0.09)	0.96 (0.09)	0.96 (0.09)	0.98 (0.10)
Smoke				
Never/Former	49 (91%)	37 (90%)	37 (90%)	14 (93%)
Current	5 (9%)	4 (10%)	4 (10%)	1 (7%)
Diabetes				
No	38 (70%)	26 (63%)	26 (63%)	15 (100%)
Yes	16 (30%)	15 (37%)	15 (37%)	10 (67%)
Hyperlipidemia				
No	33 (61%)	26 (63%)	26 (63%)	5 (33%)
Yes	21 (39%)	15 (37%)	15 (37%)	7 (47%)
Hypertension				
No	27 (50%)	21 (51%)	21 (51%)	7 (47%)
Yes	27 (50%)	20 (49%)	20 (49%)	8 (53%)

MAFLD: Metabolic-Associated Fatty Liver Disease: MAFLD1; presence of hepatic steatosis in overweight/obese subjects; MAFLD3: presence of hepatic steatosis and type 2 diabetes mellitus; SBP: Systolic Blood Pressure; DBP: Diastolic Blood Pressure; BMI: Body Max Index; TGL: Triglycerides; Chol-T: Total Cholesterol; HDL-C: High-Density Lipoprotein; LDL-C: Low-Density Lipoprotein; RC: Remnant Cholesterol; ALT: Alanine Amino Transferase; HOMA-IR: Homeostatic Model Assessment for Insulin Resistance; WHR: Waist to Hip Ratio.

**Table 3 jcm-11-04339-t003:** Joint modeling of longitudinal trajectory (BMI) and Time-to-Event (Loss of MAFLD diagnostic criteria): Effect of Adherence to LGIMD and CEP.

	MAFLD		MAFLD1		MAFLD3	
	Coeff ^#^	95%CI	Coeff	95%CI	Coeff	95%CI
Longitudinal						
Adherence LGIMD/CEP	0.04	−0.05; 0.13	0.04	−0.05; 0.13	0.04	−0.05; 0.13
Time-to-Event ^#^						
ln (BMI)	−9.66 **	−13.62; −5.69	−9.92 **	−13.83; −6.01	−4.45 **	−6.63; −2.27
Adherence LGIMD/CEP	−1.78 *	−3.40; −0.16	−1.57 **	−3.11; −0.01	−1.26 *	−2.17; −0.35
Test Overall Effect Adherence LGIMD/CEP	−2.15 *	−4.04; −0.26	−1.94 *	−3.78; −0.11	−1.43 *	−2.44; −0.41

BMI: Body Mass Index; LGIMD: Low Glycemic Index Mediterranean Diet; CEP: Combined Exercise Program; MAFLD: Metabolic-Associated Fatty Liver Disease; MAFLD1; presence of hepatic steatosis in overweight/obese subjects; MAFLD3: presence of hepatic steatosis and type 2 diabetes mellitus; ^#^ Waist to Hip Ratio; Protein C Reactive; Sex; Alanine Aminotransferase; Glutamyl Transferase; Hypertension and HOMA-IR (Homeostatic Model Assessment for Insulin Resistance) Index adjusted coefficients; 95%CI: 95% Confidence Interval; * *p*-value < 0.05; ** *p*-value < 0.001.

**Table 4 jcm-11-04339-t004:** Joint modeling of longitudinal trajectory (HOMA-IR) and Time-to-Event (Loss of MAFLD diagnostic criteria): Effect of Adherence to LGIMD and CEP.

	MAFLD		MAFLD1		MAFLD3	
	Coeff ^#^	95%CI	Coeff	95%CI	Coeff	95%CI
Longitudinal ^§^						
Adherence LGIMD/CEP	−0.27	−0.61; 0.07	−0.27	−0.61; 0.07	−0.27	−0.61; 0.07
Time-to-Event ^#^						
ln(Homa-IR)	−3.38 *	−6.69; −0.06	−0.87	−3.07; 1.32	0.58	−0.43; 1.60
Adherence LGIMD/CEP	−2.30 *	−4.25; −0.34	−1.51 *	−3.15; −0.00	−1.11 *	−2.05; −0.17
Test Overall Effect Adherence LGIMD/CEP	−1.39	−3.45; 0.59	−1.28	−2.79; 0.23	−1.26 *	−2.19; −0.34

^§^ HOMA-IR: Homeostatic Model Assessment for Insulin Resistance; MAFLD: Metabolic Associated Fatty Liver Disease; LGIMD: Low Glycemic Index Mediterranean Diet; CEP: Combined Exercise Program; MAFLD1; presence of hepatic steatosis in overweight/obese subjects; MAFLD3: presence of hepatic steatosis and type 2 diabetes mellitus; ^#^ Waist to Hip Ratio, Protein C Reactive, Sex, Alanine Aminotransferase, Glutamyl Transferase, Hypertension and Body Mass Index adjusted coefficients; 95% CI: 95% Confidence Interval; * *p*-value < 0.05.

## Data Availability

Data are available upon reasonable request from the corresponding.

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
