# Peer review of "The Effect of Low Glycemic Index Mediterranean Diet and Combined Exercise Program on Metabolic-Associated Fatty Liver Disease: A Joint Modeling Approach"

_jcm, 2022, doi:10.3390/jcm11154339_

Round 1

Reviewer 1 Report

In this manuscript, Curci and the colleagues reported the effect of low glycemic index Mediterranean diet (LGIMD) and combined exercise program (CEP) on MAFLD patients. They found that LGIMD/CEP significantly improved MAFLD status, and also found BMI and HOMA-IR were good predictors of MAFLD disease status. This study is interesting and new. I have some concerns remaining as followings.

Comments

1.         In Figure 1 and 2, the authors demonstrated the significance of LGIMD/CEP on MAFLD by hazard ratio (HR) of time-to-event. However, the authors did not fully explain the interpretation of these results precisely. The authors should describe about this figure more precisely.

2.         In addition to the above comments, the discussions of this paper are rather thin, and additional information is required. The authors should discuss about each LGIMD and CEP. How about the effect of LGIMD and CEP on each MAFLD1 and MAFLD3? Were there any differences of the effects of LGIMD and CEP on each MAFLD1 and MAFLD3?

Reviewer 2 Report

The article entitled “The effect of Low Glycemic Index Mediterranean Diet and Combined Exercise Program on Metabolic-Associated Fatty Liver Disease: a joint Modeling Approach” by Curci et al. has an interesting subject for study on the relationship between the effects of Low Glycemic Index Mediterranean Diet (LGIMD) and Combined Exercise Program (CEP) on Metabolic-Associated Fatty Liver Diseases (MAFLD).

The article has several major shortcomings, the most important of which is the wrong design of the study. The authors did NOT perform a randomization of the subjects included in the study and did NOT have a control group (control missing).

Therefore, the results obtained must be questioned.

The way of presenting the results, but especially of the conclusions is superficial and inconclusive, without concrete statistical values, percentages, etc.

In addition, the authors did not adhere to MeSH for keywords choice and have several imperfections in wording and form. I recommend that the paper be reviewed by a native English speaker to correct expression.

I believe that the article must undergo a major revision in order to be accepted, because it is NOT scientifically credible, even if the work of the authors was important!

In conclusion, the article is not written on the standards of a credible scientific article. I think this manuscript should be rewritten and redesigned to improve it and only then could it be published!

Reviewer 3 Report

Great study looking at an important topic in MAFLD. It's difficult to do study involving diet. Below are some questions:

You had chosen 10 random weeks to assess the MAI. 

1) were the subjects know which week that they will be assessed for the MAI? In other words, were they blind and only the investigators know? if the subject knows which week, then they might be more "compliant/adherent" to the prescribed diet regimen, leading to potential bias.

2) in those pts with diabetes, is their glycemic control optimal?

Author Response

Please the attachment

Round 2

Reviewer 1 Report

The atuhors well-revised their manuscrpit. I have no further comments on this mnuscript. 

Author Response

thank you for the comment

Reviewer 2 Report

Even though the authors Curci et al. tried to improve their article “The Effect of Low Glycemic Index Mediterranean Diet and Combined Exercise Program on Metabolic-Associated Fatty Liver Disease: a Joint Modeling Approach”, it cannot be published in the prestigious JCM journal in its current form. In total, the manuscript is not written on the basis of correct scientific standards. The paper must be redesigned and rewritten with the control group(s). E.g: - a diet-only control group for patients who do NOT like or cannot exercise easily; - an exercise-only control group, for patients who do NOT like to diet or can not easily follow a diet program from different reasons and for a long time; OR, a control group that does NOT want or can NOT diet or exercise. I am convinced that this manuscript must be rewritten and redesigned! THANK YOU VERY MUCH !

Author Response

We have responded to the reviewer's previous criticisms by relying on books and articles published in journals of undoubted prestige. Furthermore, our group has acquired the methodological and statistical skills sufficient to support our position over time. We are not willing to redesign the study (it would not be honest scientifically speaking) nor to rewrite the paper as we have not received any suggestions.